# Association Analyses between Single Nucleotide Polymorphisms in *ZFAT*, *FBN1*, *FAM184B* Genes and Litter Size of Xinggao Mutton Sheep

**DOI:** 10.3390/ani13233639

**Published:** 2023-11-24

**Authors:** Yiming Gong, Qiuju Chen, Xiaolong He, Xiangyu Wang, Xiaoyun He, Yunfei Wang, Zhangyuan Pan, Mingxing Chu, Ran Di

**Affiliations:** 1State Key Laboratory of Animal Biotech Breeding, Institute of Animal Science, Chinese Academy of Agricultural Sciences (CAAS), Beijing 100193, China; 82101215374@caas.com (Y.G.); xiangyu_wiggle@163.com (X.W.); hedayun@sina.cn (X.H.); zhypan01@163.com (Z.P.); 2Bayannur Institute of Agriculture & Animal Husbandry Science, Bayannur 015000, China; chenqiuju1958@163.com (Q.C.); meteor1001@163.com (Y.W.); 3Inner Mongolia Academy of Agricultural and Animal Husbandry Sciences, Hohhot 010031, China; hexiaolong1983@163.com

**Keywords:** sheep, fecundity, *FBN1*, molecular marker, MassARRAY

## Abstract

**Simple Summary:**

*FBN1*, *ZFAT* and *FAM184B* have been screened as candidate genes for the reproduction of sheep. Therefore, it is necessary to verify these genes in the population and determine the associated loci for litter size. The association of litter size with the genotypes of three candidate genes was analyzed using the fixed effects model in Xinggao mutton sheep. The results showed that the g.160338382 T > C in *FBN1* was significantly associated with litter size in Xinggao mutton sheep and that this effect was independent of the *FecB* mutation. Overall, this study provides a useful genetic marker for improving sheep fecundity.

**Abstract:**

Previous studies have screened key candidate genes for litter size in sheep, including fibrillin-1 (*FBN1*), family with sequence similarity 184 member B (*FAM184B*) and zinc finger and AT-hook domain containing (*ZFAT*). Therefore, it is necessary to verify these genes in the Xinggao mutton sheep population and determine the associated loci for litter size. In this study, three loci (*FBN1* g.160338382 T > C, *FAM184B* g.398531673 C > T and *ZFAT* g.20150315 C > T) were firstly screened based on the population differentiation coefficient between the polytocous and monotocous sheep groups. Then, population genetic analysis and association analysis were performed on these loci. The results revealed that the g.160338382 T > C in *FBN1* was significantly associated with the litter size of sheep. Moreover, there was no significant interaction effect between the g.160338382 T > C locus and *FecB* on litter size. Notably, g.160338382 T > C is adjacent to the anterior border of exon 58 and belongs to a splice polypyrimidine tract variant, which may lead to alternative splicing and ultimately cause changes in the structure and function of the protein. In summary, our results provided a potentially effective genetic marker for improving the litter size of sheep.

## 1. Introduction

Litter size is a very important trait affecting the economic benefits of the mutton sheep industry or farmers. However, the molecular mechanism of litter size in sheep has not been completely elucidated. The known major genes for litter size are very limited in sheep; therefore, the characterization of important genes and molecular markers for prolificacy is crucial for improving reproductive efficiency in sheep. By sequencing the whole genome of the 248 sheep from 36 landraces and 6 improved breeds around the world, Li et al. [1] reported key candidate genes for reproduction, including fibrillin-1 *(FBN1*), family with sequence similarity 184 member B *(FAM184B*) and zinc finger and AT-hook domain containing (*ZFAT*), combining the methods of genome-wide association analysis (GWAS) and selective signature analysis. This provides us with important candidate genes for further molecular characterization of the litter size of sheep.

*FBN1* has been proved to be associated with Marfan syndrome, polycystic ovary (POCS) and ovarian follicular development [2,3,4,5]. In recent years, it was discovered that *FBN1* may encode a novel protein, asprosin [6], which may be a novel regulator for ovarian follicular function [7]. Maylem et al. [7] verified the negative relationship between the abundance of *FBN1* mRNA and the diameters of bovine ovarian follicles. Using the GWAS method, the *FAM184B* gene has been screened and it was found that it was associated with the litter size of sheep, and was also highly expressed in the sheep follicle and hypothalamus [8]. Similarly, the gene had also been considered as a candidate gene for reproductive traits in Jinghai yellow chicken [9]. The *ZFAT* gene, strongly expressed in the placenta, was labeled as a new imprinted gene [10]. The expression of this gene is down-regulated in the placenta during pregnancy, especially in pathological placenta [10]. Kobayashi [11] found that the *ZFAT* gene was highly correlated with reproductive diseases such as endometriosis. However, the association between the above three genes (*FBN1*, *FAM184B*, *ZFAT*) and the litter size of sheep has not been studied.

Therefore, in this study, these three genes were selected as candidate genes to analyze their association with the litter size of polytocous and monotocous sheep. Firstly, based on our previous resequencing data of 10 sheep breeds [12], the polymorphic loci in these three genes were screened between the polytocous sheep group (Cele black sheep, Small Tail Han sheep, Hu sheep and Australian Merino sheep) and the monotocous sheep group (Prairie Tibetan sheep, Vally Tibetan sheep, Euler sheep, Bayanbulak sheep, Ujumqin sheep and Tan sheep). Then, the allele frequencies of each polymorphic locus in the 10 breeds and the population differentiation coefficient (*F_st_*) of each locus between the two groups were calculated. Next, the key candidate loci (g.160338382 T > C in *FBN1*, g.398531673 C > T in *FAM184B* and g.20150315 C > T in *ZFAT*) were selected for association analysis and if they met any one of the conditions: ① non-synonymous mutations with *F_st_* > 0.05, ② SNPs with *F_st_* > 0.15 in intron. *FecB* mutation is prevalent in many sheep breeds in China, and its effect is strong. In order to avoid *FecB* masking the effects of other genes and to analyze whether the effect of candidate loci is independent of *FecB* mutation, we analyzed the interaction effect between candidate loci and *FecB* on the litter size of ewes. This study will help to seek new genes and genetic markers associated with litter size for sheep breeding.

## 2. Materials and Methods

### 2.1. Animal Preparation and Sample Collection

Blood samples were collected from 751 ewes, which are from three polytocous breeds (Hu sheep, Cele black sheep, Xinggao mutton sheep) and two monotocous breeds (Sunite sheep, Bamei mutton sheep) (Table 1). Xinggao mutton sheep is a new breed with high fertility and meat performance. Hu sheep and Cele black sheep are famous polytocous breeds in China. Both Sunite and Bamei mutton sheep are famous for high rate of meat production in China; however, they are monotocous breeds. DNA were extracted using TIANamp Genomic DNA kit (TIANGEN Co., Ltd., Beijing, China. DP304-03). The litter size information of Xinggao mutton sheep with the second and third parities was recorded.

### 2.2. Genotyping

The genotyping for the first three SNPs was conducted in a MassARRAY^®^ SNP system. The genomic DNA of the samples was amplified using the amplification primers shown in Table 2. Then, the amplified products were digested with SAP enzyme. After that, the digested products were used as templates for extension reaction using the EXT primers (Table 2). The single-base extended primers for three loci were designed via MassARRAY Assay Design v. 3.1 based on the sheep sequences of *FBN1, FAM184B* and *ZFAT* available in GenBank ARS-UI_Ramb_v2.0 (accession no.: NC_056062.1, NC_056060.1; NC_056059.1). Finally, the extended products were analyzed by matrix-assisted laser desorption ionization time-of-flight mass spectrometry to determine the genotypes of the SNP loci. Detailed information about the system and procedures has previously been described [13,14]. *FecB* was amplified and genotyped according to our previous study [15].

### 2.3. Statistical Analysis

Allele and genotype frequency, polymorphism information content (*PIC*), heterozygosity (*He*) and number of effective alleles (*Ne*) were calculated using the following formulae:
(1)
$$PIC=1-\sum_{i=1}^{n}{p_{i}}^{2}-\sum_{i=1}^{n-1}\sum_{j=i+1}^{n}2{p_{i}}^{2}{p_{j}}^{2}$$


(2)
$$He=1-\sum_{i=1}^{n}{p_{i}}^{2}$$


(3)
$$Ne=1/\sum_{i=1}^{n}{p_{i}}^{2}$$

where *n* is the number of alleles, p*_i_* is the allele frequency of the *i*th allele and *p_j_* is the allele frequency of the *j*th allele.

Chi-square test was used to detect whether the genotype distribution of each locus deviated from Hardy–Weinberg equilibrium. For the data that cannot be tested by chi-square test, we used the method of Fisher’s exact test to calculate the *P* value. The association of litter size with the genotypes of four SNPs was analyzed by SPSS 26 (ANOVA). The association of litter size with *FBN1* and *FecB* was analyzed using the following fixed effects model, with least squares means used for multiple comparisons of litter size among the different genotypes in Xinggao mutton sheep: *y* =*μ* + *P* + *G1* + *G2* + *G1G2* + *e*, where *y* is the phenotypic value of litter size, *μ* is the population mean, *P* is the fixed parity effect (two levels), *G1* is the fixed effect for the candidate SNPs, *G2* is the fixed effect for *FecB*, *G1G2* is the fixed interaction effect for the candidate SNPs and *FecB* and *e* is the random error effect of each observation. Analysis was performed using MASS package in R software (aov, Version 4.0.3) [16].

### 2.4. Protein Structure and Interaction Network Analysis

For the SNP significantly related to litter size, we further analyzed the location of the SNP and its influence on protein structure. Then, interaction network where the protein is located was constructed. If there is no existing protein network in sheep, it is necessary to construct a phylogenetic tree to determine the species whose protein network can be referenced. The candidate protein sequences of 11 species were aligned by CLUSTALW (https://www.genome.jp/tools-bin/clustalw, accessed on 20 September 2023) and displayed by ESPript 3.0 (https://espript.ibcp.fr/ESPript/cgi-bin/ESPript.cgi, accessed on 20 September 2023), and phylogenetic tree was constructed through the MEGA 11 and beautified on iToL (https://itol.embl.de/, accessed on 20 September 2023). The secondary structure of candidate protein was predicted using PredictProtein. The interaction network for candidate protein in model animals was predicted via STRING database v.12.0 [17] (https://cn.string-db.org/, accessed on 20 September 2023) and BioGRID database v.4.4 [18] (https://thebiogrid.org/, accessed on 20 September 2023), respectively.

## 3. Results

### 3.1. Genotyping and Population Genetic Analysis of Candidate SNPs in FBN1, FAM184B and ZFAT Genes

The candidate SNPs (g.160338382 T > C in *FBN1*, g.398531673 C > T in *FAM184B*, g.20150315 C > T in *ZFAT*) were genotyped in all ewe samples, and the genotyping results are shown in Figure 1. The results of population genetic analysis (Table 3) indicated that the g.160338382 locus had moderate polymorphism (0.25 < *PIC* < 0.5) in most breeds except for Cele black sheep, the g.398531673 C > T locus had moderate polymorphism in Cele black sheep (0.25 < *PIC* < 0.5) and g.20150315 C > T had moderate polymorphism in Xinggao mutton sheep. The g.29382188 T > C locus had moderate polymorphism in Xinggao mutton sheep and Cele sheep (0.25 < *PIC* < 0.5). Hardy–Weinberg equilibrium tests revealed that g.160338382 T > C was under the Hardy–Weinberg equilibrium in five populations (*p* > 0.05), but the g.398531673 C > T locus was deviated from the Hardy–Weinberg equilibrium in Xinggao mutton sheep, Sunite sheep and Cele black sheep (*p* < 0.05). 

### 3.2. Associations Analysis between loci in FBN1, FAM184B, ZFAT, BMPR1B and Litter Size of Xinggao Mutton Sheep

The results of association analysis revealed that the g.160338382 T > C locus in *FBN1* was significantly associated with litter size in Xinggao mutton sheep (Table 4). For this locus, the litter size in ewes with the CC genotype was significantly lower than that in ewes with the TT genotype in both the second parity and the third parity (*p* < 0.05). No significant association was observed between litter size and two loci in *FAM184B* and *ZFAT* genes. For the *FecB* mutation in *BMPR1B*, the litter sizes of ewes with the GG genotypes were significantly higher than those of ewes with the AA genotype in both parities (*p* < 0.05). However, there was no significant interaction effect between *FecB* and the g.160338382 T > C locus on litter size (Table 5, *Pr* > 0.05). Additionally, the litter size of Xinggao mutton sheep was significantly influenced by parity (*p* = 0.000019).

### 3.3. Protein Structure and Interaction Network Analysis on FBN1

Based on the results of association analysis, which showed that only *FBN1* g.160338382 T > C (except for *FecB*) had a significant relationship with litter size, we further checked the protein structure of FBN1 and analyzed the effect of this mutation on protein structure (Figure 2A). Interestingly, SNP g.160338382 T > C is adjacent to the anterior border of exon 58 (Figure 2B) and belongs to a splice polypyrimidine tract variant, which may lead to alternative splicing and ultimately cause changes in the structure and function of the protein. As shown in Figure 2A, exon 58 is located in the EGF-like domain, whose release is essential for inducing the ovulation process [19]. This suggests that the mutation might be closely related to ovulation.

In order to further understand the biological function of FBN1 in reproduction, we expect to obtain some valuable information from the protein interaction network. Since there is not an existing protein network for FBN1 protein in sheep, we first constructed a protein evolutionary tree (Figure 3A) to select the species that can be used for reference in protein network analysis. The result of the protein evolutionary tree displayed that two model animals (*Capra hircus* and *Bos taurus*) had the closest protein evolutionary relationship with sheep. In addition, *Homo sapiens* would also been involved in the protein interaction network analysis because there are many similarities in the reproduction mechanism between humans and sheep (bicorned uterus and similar ovulation rate) and in-depth function studies have been carried out on FBN1 protein in *Homo sapiens* [20,21]. Protein interaction network analysis for *Capra hircus, Homo sapiens* and *Bos taurus* demonstrated the 10 proteins most closely interacting with FBN1 (Figure 3B–D), and, among them, the five proteins (ELN, LUM, MFAP5, COL1A2, COL5A2) were associated with the proliferation of ovarian granulosa cells and cell markers for theca interna and granulosa cells [22,23,24]. Otherwise, the predicted protein network map of human FBN1 provided by the BioGRID database showed that KIAA1429, which is related with follicular development, has the high correlation with FBN1 besides the above interacting proteins (Figure 3E). These findings indicate that FBN1 plays an important role in follicular development together with the interacting proteins in these species.

## 4. Discussion

By sequencing the whole genome of the 248 local sheep from 36 landraces and 6 improved breeds around the world, Li et al. [1] screened key candidate genes for reproduction, including *FBN1*, *FAM184B* and *ZFAT*, combining the methods of genome-wide association analysis (GWAS) and selective signature analysis. However, subsequent association analyses of these three genes with litter size have never been performed. Therefore, in this study, we first screened for polymorphic loci in these three genes based on our previous resequencing data of 10 sheep breeds and then determined the locus associated with litter size. 

Population genetic analysis indicated that the locus in *FAM184B* is not under the Hardy–Weinberg equilibrium in four breeds, including Xinggao mutton sheep, Cele black sheep, Sunite sheep and Bamei black sheep, suggesting that the locus may be subject to natural or artificial selection. In previous studies, the *FAM184B* gene was found in a candidate QTL region (close to Chr6:37.53 Mb) for meat traits in sheep and its eight SNPs have been identified as significant pleiotropic SNPs linked with meat traits [25,26]. Therefore, the *FAM184B* gene might be strongly selected during the domestication of the mutton sheep population. On the other hand, the other two loci of *FBN1* and *ZFAT* were both under the Hardy–Weinberg equilibrium in five breeds, so these two loci seem to have not been subjected to strong artificial selection. Moreover, as a famous polytocous locus, g.29382188 T > C in *BMPR1B* was deviated from the Hardy–Weinberg equilibrium in Hu sheep. This implies that this locus was subject to artificial selection during the domestication of Hu sheep in order to increase the litter size.

In this study, we analyzed the association between four loci and litter size with different parities. The results showed that *FBN1* g.160338382 T > C and *FecB* had a significant association with a litter size of second and third parity in Xinggao mutton sheep. As a major mutation of the high prolificacy of sheep, *FecB* was present in some Chinese prolific breeds of sheep [27], such as Hu sheep, Small Tail Han, Cele and Duolang sheep. In this study, we also found that *FecB* mutation was significantly correlated with litter size, but there was no significant interaction effect between *BMPR1B* and the *FBN1* g.160338382 T > C. This implies that the effect of *FBN1* g.160338382 T > C is obvious and independent of *FecB* on litter size.

FBN1, a 350-KD glycoprotein, is the main component of microfibrils in the extracellular matrix [28]. As a member of the fibrillin protein family, FBN1 plays an indispensable role in fibrosis by regulating TGF-β signaling in the extracellular matrix [29]. Previous studies showed that fibroblasts or stromal cells are the major cell type present in the stroma of many organs [30] and can result in defects in the function of organs, such as polycystic ovary syndrome (POCS) [31]. By measuring the mRNA abundance of FBN1 protein in the granulosa cell of *Bovine* and *Buffalo* ovaries, the researchers found that, in both species, the *FBN1* mRNA abundance in small follicles was significantly greater than that in large follicles [7,32]. Furthermore, FBN1 plays an important role in follicular development [5] and granulosa cell proliferation [32].

Interestingly, as an intron mutation, *FBN1* g.160338382 T > C might affect the splicing and transcription of exons 58 of FBN1. As a polypeptide protein translated from 65 exons, FBN1 protein is encoded by 2876 amino acids. Among them, amino acids 1–24 encode signal peptides and amino acids 25–2876 encode protein binding (EGF-like domain (EGF-like calcium-binding domain)), DNA binding, RNA binding and a small segment of a disordered region. Exon 58 (ENSOARE00000208277) is encoded by 42 amino acids and is located in the EGF-like domain, including DNA binding and RNA binding. Previous studies showed that the release of the EGF-like domain by a specific enzyme is essential for interaction with the EGFR for granulosa cell luteinization, cumulus expansion and oocyte maturation [19,33]. EGF-like domains of exon 58 are 45–60-amino-acid-long domains located between the transmembrane domains and the mucin domains. In recent years, researchers have found that adding EGF to cumulus cells cultured in vitro can phosphorylate the receptors (ERK1/2, EGFR) in the cells and activate the expansion and the maturation of oocytes of *Sus scrofa* [34], *Mouse* [35] and *Bovine* ovaries [36]. During ovulation, multiple signaling pathways in cumulus cells and granulosa cells are activated. EGF-like factors are responsible for the RAS-cRAF-MEK1-ERK1/2 pathway in spermatogonia and granulosa cells [37,38]. Previous studies have shown that the addition of EGFR inhibitors or specific knockout of EGF domain coding genes in follicular cells can cause the expansion of the cumulus to become dramatically blocked [39,40,41]. They also clearly revealed that the release of the EGF domain acted on EGFR in both granulosa cells and cumulus cells to activate the ERK1/2 pathway [42]. Coincidentally, this mutation is adjacent to the anterior border of exon 58 (Figure 2B) and belongs to a splice polypyrimidine tract variant, which may lead to alternative splicing and changes in the structure of protein and may ultimately affect the proliferation and cumulus expansion of granulosa cells and ovulation.

In addition, FBN1 protein had been shown to play an important role in cell differentiation and apoptosis in different cell types and species [5,43]. FBN1 stimulates follicular atresia in the presence of BMP15 (a ligand that can stimulate follicular development) based on its effects on cumulus cell apoptosis. In the presence of BMP15, FBN1 depletion protected cumulus cells from apoptosis. Similarly, when *FBN1* was depleted, the addition of BMP15 played a protective role for cumulus cells [44]. Thus, we summarized that FBN1 is an important regulatory factor for BMP15 to inhibit the apoptosis of porcine ovarian cumulus cells. When the mutation causes abnormal transcription, it will promote the apoptosis of granulosa cells and reduce the number of ovulations. Therefore, this may be the reason why we observed a decrease in litter size in mutant ewes in this study.

The results of the protein interaction network showed that the six proteins that have the closest relationship with FBN1 are associated with follicular development and ovulation. For interacting proteins KIAA1249, previous studies have shown that the specific deletion of KIAA1249 in mouse oocytes leads to defects in follicular development, loss of the meiotic performance of oocytes and ultimately female infertility [45]. The five proteins (ELN, LUM, MFAP5, COL1A2 and COL5A2) were highlighted from the protein network provided by the STRING. ELN was identified as a gene significantly related to chicken follicular development by transcriptome profiling analysis [46]. LUM has been reported to be involved in ovulation in *Homo sapiens* and bovine [47,48]. MFAP5 was related to ovarian hyperplasia and endometrial growth [49,50]. As the main component of extracellular matrix (ECM) protein in the mammalian ovary, COL1A2 plays a vital role in follicular development and corpus luteum formation [22,51,52]. In summary, the FBN1 protein forms a network with the above proteins to coordinately regulate follicular development and ovulation.

## 5. Conclusions

The present study revealed that an SNP (g.160338382 T > C) in *FBN1* is significantly associated with litter size in Xinggao mutton sheep. Moreover, this effect is independent of the *FecB* mutation. Therefore, the locus might be considered as a potentially effective genetic marker for improving litter size.

## Figures and Tables

**Figure 1 animals-13-03639-f001:**
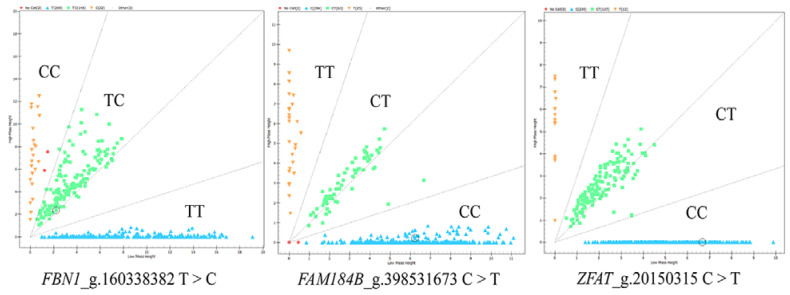
Genotyping results of candidate SNPs in *FBN1*, *FAM184B*, *ZFAT* genes using MassARRAY^®^ SNP system.

**Figure 2 animals-13-03639-f002:**
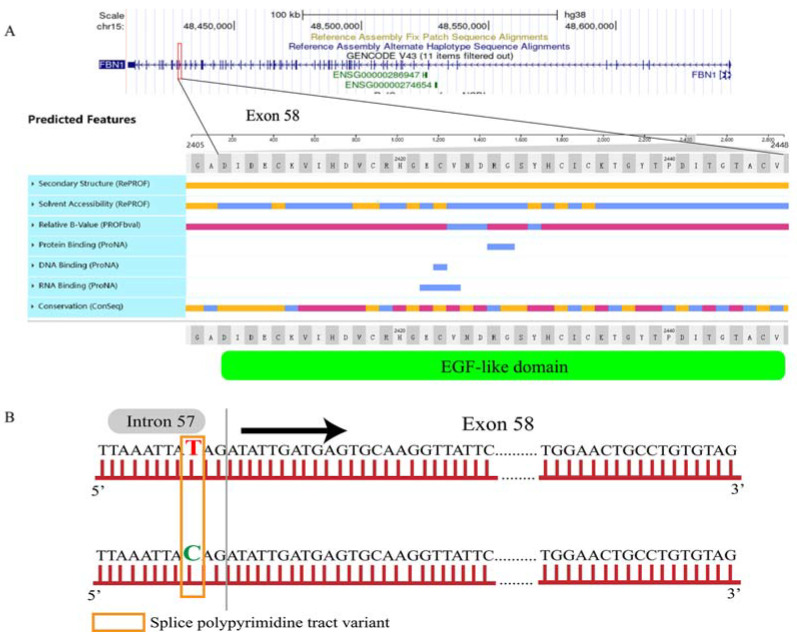
The sequence and predicted features of exon 58 in sheep FBN1 displayed using PredictProtein (**A**). The position of SNP g.160338382 T > C is adjacent to anterior border of exon 58 in sheep (**B**).

**Figure 3 animals-13-03639-f003:**
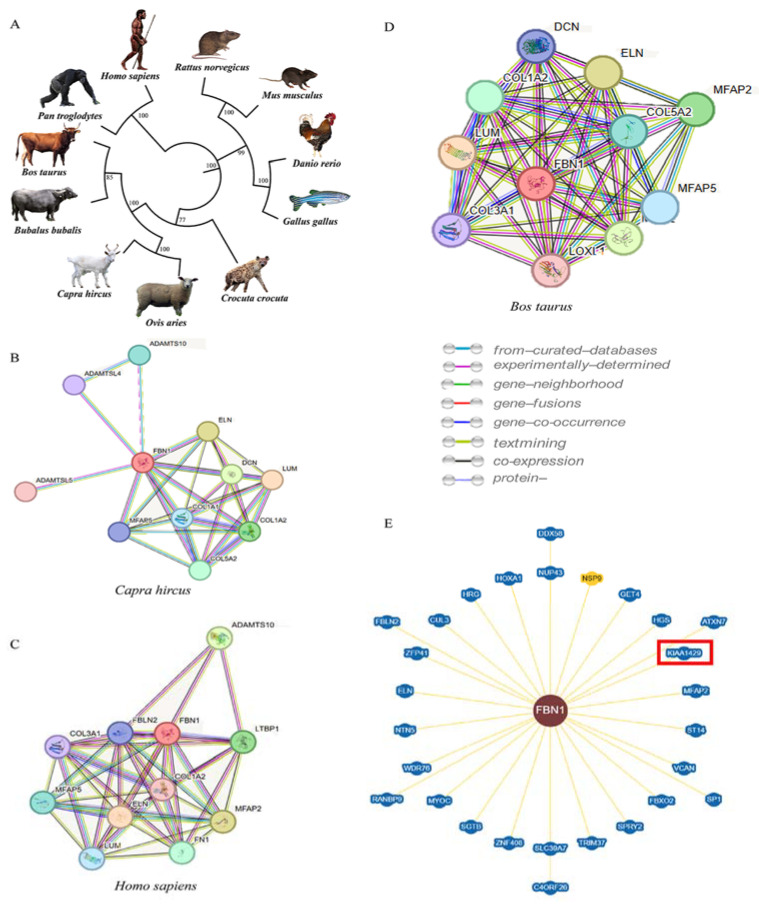
Phylogenetic tree of FBN1 protein in 11 species (**A**). Interaction networks of proteins for FBN1 in *Capra hircus* (**B**), *Homo Sapiens* (**C**) and *Bos tauru*s (**D**) as provided by String database (Version 12). Interaction networks of proteins for FBN1 in Homo species provided by BioGRID database (**E**). The KIAA1429 protein was in the red square.

**Table 1 animals-13-03639-t001:** Information on ewes used in this study.

Breeds	Number	Fertility	Sampling Location
Hu sheep	96	Polytocous	Xuzhou, Jiangsu Province, China
Cele black sheep	96	Polytocous	Cele, Xinjiang Uygur Autonomous Region, China
Xinggao mutton sheep	367 (with litter size)	Polytocous	Xing’an league, Inner Mongolia Autonomous Region, China
Sunite sheep	96	Monotocous	Bayannaoer, Inner Mongolia Autonomous Region, China
Bamei mutton sheep	96	Monotocous	Bayannaoer, Inner Mongolia Autonomous Region, China

**Table 2 animals-13-03639-t002:** The primer sequences for MassARRAY analysis.

Loci	Primer Sequence (5′-3′)
g.160338382	F-ACGTTGGATGTCCCCATGTCGGCAAACATC
	R-ACGTTGGATGCCAGTAGTACATATTGACCC
	EXT-CCTTGCACTCATCAATATCT
g.398531673	F-ACGTTGGATGCCTCCTTGAGCTGTGAGTTC
	R-ACGTTGGATGTCCCAAAGGCAAGAGTGCAG
	EXT-GGCAAGGAGGCAGCATCCCT
g.20150315	F-ACGTTGGATGACCAAGTACCAGGCGCTGGA
	R-ACGTTGGATGTGGAGCTGACGAACTTCTTG
	EXT-ACCCGGCGCTGGAGCTGCACGTC

**Table 3 animals-13-03639-t003:** Population genetic analysis of candidate SNPs in five sheep breeds.

Genes	SNPs	Breeds	Genotype Frequency			Allele Frequency		*PIC*	*He*	*Ne*	*Chi*-Square Test (*p*-Value)
*FBN1*	Chr7: g.160338382 T > C		**CC**	**TC**	**TT**	**C**	**T**				
		Xinggao mutton sheep	0.06	0.40	0.55	0.26	0.74	0.31	0.38	1.61	0.44
		Hu sheep	0.19	0.47	0.34	0.43	0.57	0.37	0.49	1.96	0.76
		Cele sheep	0.02	0.21	0.77	0.13	0.88	0.19	0.22	1.28	0.64
		Sunite sheep	0.03	0.38	0.59	0.22	0.78	0.29	0.34	1.53	0.33
		Bamei mutton sheep	0.34	0.50	0.16	0.59	0.41	0.37	0.48	1.93	0.72
*FAM184B*	Chr6: g.398531673 C > T		**CC**	**CT**	**TT**	**C**	**T**				
		Xinggao mutton sheep	0.77	0.16	0.07	0.85	0.15	0.22	0.25	1.34	0.00
		Hu sheep	0.89	0.10	0.01	0.94	0.06	0.11	0.12	1.13	0.28
		Cele sheep	0.54	0.32	0.14	0.70	0.30	0.33	0.42	1.72	0.02
		Sunite sheep	0.81	0.14	0.05	0.88	0.12	0.19	0.21	1.27	0.00
		Bamei mutton sheep	0.99	0.01	0.00	0.99	0.01	0.01	0.01	1.01	NA
*ZFAT*	Chr9: g.20150315 C > T		**CC**	**CT**	**TT**	**C**	**T**				
		Xinggao mutton sheep	0.61	0.36	0.03	0.79	0.21	0.28	0.33	1.50	0.19
		Hu sheep	0.96	0.04	0.00	0.98	0.02	0.04	0.04	1.04	0.83
		Cele sheep	1.00	0.00	0.00	1.00	0.00	0.00	0.00	1.00	NA
		Sunite sheep	0.98	0.02	0.00	0.99	0.01	0.02	0.02	1.02	NA
		Bamei mutton sheep	1.00	0.00	0.00	1.00	0.00	0.00	0.00	1.00	NA
*BMPR1B*	Chr6: g.29382188 T > C		**TT**	**TC**	**CC**	**T**	**C**				
		Xinggao mutton sheep	0.14	0.48	0.38	0.38	0.62	0.36	0.47	1.88	0.67
		Hu sheep	0.01	0.06	0.93	0.04	0.96	0.08	0.09	1.09	0.03
		Cele sheep	0.42	0.52	0.06	0.68	0.32	0.34	0.44	1.77	0.06
		Sunite sheep	0.81	0.18	0.01	0.90	0.10	0.16	0.18	1.22	0.95
		Baimei mutton sheep	0.99	0.01	0.00	1.00	0.00	0.01	0.01	1.01	NA

**Table 4 animals-13-03639-t004:** Least squares mean and standard deviation of litter size for different genotypes in Xinggao mutton sheep.

Genes	SNPs	Genotype	Litter Size (Mean ± SD)	
			Second Parity (N)	Third Parity (N)
*FBN1*	g.160338382 T > C	CC	1.85 ± 0.555(13) ^b^	2.20 ± 0.837(5) ^b^
		TC	2.32 ± 0.877(103) ^ab^	2.83 ± 1.124(35) ^ab^
		TT	2.70 ± 0.789(133) ^a^	3.15 ± 0.931(55) ^a^
*FAM184B*	g.398531673 C > T	CC	2.43 ± 0.914(208)	2.89 ± 1.090(71)
		CT	2.53 ± 0.869(45)	3.05 ± 0.887(20)
		TT	2.36 ± 0.842(14)	3.00 ± 1.225(5)
*ZFAT*	g.20150315 C > T	CC	2.40 ± 0.888(173)	2.87 ± 0.919(67)
		CT	2.53 ± 0.918(89)	3.08 ± 1.324(26)
		TT	2.71 ± 1.113(7)	3.25 ± 1.258(4)
*BMPR1B*	g.29382188 T > C	TT	1.95 ± 0.664(37) ^c^	2.46 ± 0.660(13) ^b^
		TC	2.24 ± 0.849(124) ^b^	2.60 ± 0.849(43) ^ab^
		CC	2.82 ± 0.825(95) ^a^	3.29 ± 1.071(31) ^a^

Note: Different letters for litter sizes of ewes with different genotypes indicate significant differences (*p* ≤ 0.05).

**Table 5 animals-13-03639-t005:** Effect analysis of interaction between *FecB* and the SNP (g.160338382) in *FBN1* on litter size in Xinggao mutton sheep.

Parity	SNPs		Df	Sum sq	Mean Sq	F Value	Pr (>F)
Second Parity	g.160338382	*FecB*	4	2.16	0.540	0.713	0.584
Third Parity	g.160338382	*FecB*	3	1.06	0.354	0.337	0.798

## Data Availability

The data presented in this study are available in article.

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
