# Peer review of "Association Analyses between Single Nucleotide Polymorphisms in ZFAT, FBN1, FAM184B Genes and Litter Size of Xinggao Mutton Sheep"

_animals, 2023, doi:10.3390/ani13233639_

Round 1

Reviewer 1 Report

Comments and Suggestions for Authors

The paper presents a valuable contribution to the field of animal genetics, particularly to our understanding of the genetic determinants of fecundity in sheep. By examining the candidate genes, FBN1, FAM184B, and ZFAT, and identifying the g.160338382 T > C locus in FBN1 as significantly associated with litter size, this study offers important insights for sheep breeding programs.

The manuscript is  presented in a well-structured manner, and its conclusions are consistent with the data presented. It does, however, need some minor revisions.

 I would suggest the authors clarify the following points:

 1) Why was the interaction effect between locus g.160338382 T > C in FBN1 and BMPR1B genes (FecB mutation) analyzed? It is unclear from the introduction and should be explained.

 2) Why was the Hardy-Weinberg equilibrium for the BMPR1B gene calculated for one breed only?

 3) Why were only Xinggao mutton sheep used for association analyses? Was it because only for them phenotype information about parities was recorded? Then it should be explained in Table 1.

 4) In Table 3, the p-value and effects (beta) should be reported.

 5) What do the letters a,b,c in Table 3 mean? The note says, “Different letters (a, b, c) for litter size indicate significant differences (P < 0.05).” Then why were three letters used for one value?

 6) Page 6, L11-12: Which formula was used for this statement?

 7) KIAA1429 should be marked in figure 3E, as it is referred to on page 7, lines 45–48.

  8) The order of branching of the phylogenetic tree of the FBN1 protein in 11 species shown in Fig. 3A contradicts the consensus tree of mammals. Laurasiatheria (artiodactyls and carnivores) are a sister group to all Euarchontoglires (primates and rodents), but not to primates alone. I would suggest the authors mention and explain this contradiction.

Author Response

Dear reviewer,

Thank you very much for your comments and professional advice! These opinions will help us to improve academic rigor of our article! Based on your suggestions, we have revised the manuscript carefully.Please see the attachment.

Reviewer 2 Report

Comments and Suggestions for Authors

Please find my comments in the attached review report to help you improve on the quality of the manuscript.

Comments on the Quality of English Language

The quality of English needs some improvement and I have offered suggestions in the attached review report to help the authors do this.

Author Response

Dear reviewer,

Thank you very much for your comments and professional advice! These opinions will help us to improve academic rigor of our article! Based on your suggestion, we have revised the manuscript carefully. Please see the attachment.
